# Utilizing methylglyoxal and D-lactate in urine to evaluate saikosaponin C treatment in mice with accelerated nephrotoxic serum nephritis

**Chia-Yu Lin**[1,2]**, Jen-Ai Lee**[1]**, Po-Yeh Lin**[1]**, Shih-Chun Hua**[1,3]**, Pei-Yun Tsai**[1,4]**, Bi-Li Chen**[1,2]**, Chia-En Lin**[1]**, Tzong-Huei Lee**[5]***, **Shih-Ming Chen** [1]*

**1** School of Pharmacy, Taipei Medical University, Taipei, Taiwan, **2** Department of Pharmacy, Taipei Medical University Hospital, Taipei, Taiwan, **3** Department of Pharmacy, Cathay General Hospital, Taipei, Taiwan, **4** Department of Pharmacy, WanFang Hospital, Taipei, Taiwan, **5** Institute of Fisheries Science, National Taiwan University, Taipei, Taiwan

* smchen@tmu.edu.tw (SMC); thlee1@ntu.edu.tw (THL)

**Data Availability Statement:** All relevant data are within the paper and its Supporting Information files.

## Abstract

The relationship between methylglyoxal (MGO) and D-lactate during saikosaponin C (SSC) treatment of mice with accelerated nephrotoxic serum (NTS) nephritis was investigated. NTS nephritis was induced by administration of anti-basement membrane antibodies to C57BL/6 mice and three dosages of SSC were administered for 14 days. Proteinuria, blood urea nitrogen, serum creatinine, renal histology, urinary MGO and D-lactate changes were examined. Compared to the NTS control group, the middle dosage (10 mg/kg/day) of SSC significantly alleviated the development of nephritis based on urine protein measurements ($34.40 \pm 6.85$ vs. $17.33 \pm 4.79$ mg/day, $p < 0.05$). Pathological observation of the glomerular basement membrane (GBM) revealed monocyte infiltration, hypertrophy, and crescents were alleviated, and injury scoring also showed improved efficacy for the middle dose of SSC during nephritis ($7.92 \pm 1.37$ vs. $3.50 \pm 1.14$, $p < 0.05$). Moreover, the significant decreases in urinary levels of MGO ($24.71 \pm 3.46$ vs. $16.72 \pm 2.36$ µg/mg, $p < 0.05$) and D-lactate ($0.31 \pm 0.04$ vs. $0.23 \pm 0.02$ µmol/mg, $p < 0.05$) were consistent with the biochemical and pathological examinations. This study demonstrates that MGO and D-lactate may reflect the extent of damage and the efficacy of SSC in NTS nephritis; further studies are required to enable clinical application.

## Introduction

Nephrotoxic serum (NTS) nephritis is an immune-related renal disease model using rabbits [1], rats [2] or mice [3] to simulate human autoimmune glomerulonephritis via administration of heterologous anti-glomerular basement membrane (anti-GBM) antibody [4]. First, the heterologous anti-host GBM antibodies recognize and bind to host GBM, triggering exfoliation of glomerular endothelial cells [5], platelet aggregation [6], complement activation and deposition [7], and phagocyte infiltration and proteinuria [8]. Next, the host produces anti-heterologous immunoglobulin (IgG) antibodies, which form an immune complex on the GBM and

**Funding:** The funders had no role in study design, data collection and analysis, decision to publish, or preparation of the manuscript. This study was supported by Cathay General Hospital (108CGH-TMU-06).

**Competing interests:** The authors have declared that no competing interests exist.

induce a nephritis response. Mesangial cells proliferate while monocytes infiltrate the mesangium and subendothelial region [9]. Irreversible damage occurs in the glomerulus as a result of endothelial cell proliferation, thrombus formation, thickening of the glomerular microvessels, and formation of crescents around the glomerulus [10]. Furthermore, the accelerated NTS model may simultaneously induce primary and secondary responses [11]. Clinically, glucocorticoids have been applied as a common treatment for immune-related glomerulonephritis, despite their significant side effects [12].

*Bupleuri Radix* is the dried root of some perennial herbaceous plants in the family *Umbelliferae*. Common sources are *Bupleurum Chinense* DE CANDOLLE and *Bupleurum scorzoneraefolium* WILLD [13]. *Bupleuri Radix* has multiple medicinal effects including antidepressant [14], antitumor [15], antiviral [15, 16], immune-regulating [15, 17], nephritis-inhibitory [13], anti-inflammatory [18], liver dysfunction improvement [19], and steroid-like effects [20]. These effects are predominantly due to secondary metabolites, especially saikosaponins [21]. Furthermore, it was reported that saikosaponin A, C and D may alleviate proteinuria and renal damage in GBM nephritis [22] and NTS nephritis [13].

Methylglyoxal (MGO), a highly active compound, is produced from glycolysis [23, 24] as well as lipid [25, 26] and amino acid [27] metabolism. Complications of diabetes mellitus have been widely associated with MGO-induced protein and DNA modifications and formation of advanced glycation end-products (AGEs) [26, 28–30]. In addition, MGO is related to renal injury, as observed in nephritis induced by aristolochic acid [31, 32], gentamicin [33, 34] or lead [35].

D-Lactate, an enantiomer of L-lactate, is much less prevalent in mammals than the L-form. In humans, D-Lactate originates from direct ingestion, production from intestinal bacteria, or endogenous production from MGO metabolism via the glyoxalase pathway [26, 36]. D-$\alpha$-hydroxy acid dehydrogenase may metabolize D-lactate to pyruvate, but at a much lower rate (20%) than L-lactate dehydrogenase [37]. Therefore, D-lactate is presumed to accumulate in the tissues and be excreted in urine. Compared with MGO, urinary D-lactate is relatively stable and suitable for development as an indicator of kidney injury. Previous studies showed that urinary D-lactate levels reflected the nephrotoxicity in rodents induced by aristolochic acid [38, 39], streptozotocin [40, 41], or lead [35]. However, the relationship between MGO and D-lactate in the treatment of NTS nephritis is unclear.

This study investigated the effect of different dosages of SSC on mice with accelerated NTS nephritis, and the relationship between urinary MGO and D-lactate during SSC treatment of NTS nephritis.

## Materials and methods

### Animals

C57BL/6 mice (6-weeks-old) were purchased from the National Laboratory Animal Breeding and Research Center (Taipei, Taiwan). The mice were divided into six groups ($n = 6$) and housed in a facility with a 12-h light/dark cycle and free access to water and chow (Fwusow Co. Ltd., Taichung, Taiwan). Mice were acclimatized for one week prior to the study. All animal experiments were approved by Taipei Medical University Institutional Animal Care and Use Committee (LAC-2015-0239).

### Chemicals

2,2'-Dipyridyl disulfate, 4-nitro-7-piperazino-2,1,3-benzoxadiazole, 5,6-diamino-2,4-hydroxy-pyrimidine sulfate, pentobarbital sodium, and triphenyl phosphine were purchased from Tokyo Kasei Kogyo (Tokyo, Japan). Ammonia was obtained from Showa Chemical (Tokyo,

Japan), and ammonium chloride was purchased from Kanto Chemical (Tokyo, Japan). A bicinchonic acid (BCA) Protein Assay kit was obtained from Thermo Fisher Scientific (Waltham, USA). Citric acid monohydrate and sodium hydroxide were purchased from Nacalai Tesque (Kyoto, Japan). Lithium D-lactate, lithium L-lactate, paraformaldehyde, and methylglyoxal were obtained from Sigma Chemical (St. Louis, USA). High-performance liquid chromatography (HPLC)-grade acetonitrile was obtained from Merck (Darmstade, Germany). HPLC-grade methanol and propionic acid were purchased from Mallinckrodt Baker (Phillipsburg, USA). Saikosaponin C was obtained from Wako Pure Chemical (Osaka, Japan). Triflouroacetic acid was purchased from Riedel-de Haën (Seelze, Germany).

## Purification of GBM antigen (GBM-Ag)

GBM-Ag was produced following a previously published process [11]. Briefly, the glomerular fraction was obtained from the kidneys of normal C57BL/6 mice through sieving (no. 100 and 250 meshes) and sonication. A mixture comprising trypsin, GBM-rich fraction (25 mg trypsin/10 g GBM), and 0.1 M Tris–HCl buffer containing 0.02 M $CaCl_2$ (pH 8.2) was incubated with stirring at 37˚C for 18 h. The mixture was then centrifuged (10,000 rpm at 4˚C for 60 min), and the resulting supernatant contained the crude GBM antigen (GBM-Ag).

## Preparation of immunoglobulin G (IgG)

Normal rabbit IgG was purified from the sera of NZW rabbits using an affinity chromatography kit (Mab Trap G2; Pharmacia, Sweden).

## Preparation of nephrotoxic serum (NTS)

NZW rabbits were immunized with 5 mg GBM-Ag in complete Freund's adjuvant (CFA; Sigma-Aldrich. Inc., MO, USA) at weekly intervals for five doses, and then injected with 5 mg GBM-Ag without CFA each day for 7 days. The antisera were collected 7 days later as NTS for the study.

## Study design

The experimental design (Fig 1) was modified from a previous study [13]. C57BL/6 mice (6-weeks-old) were divided into six groups ($n$ = 6) designated normal (N), SSC control (SC), nephrotoxic serum control (NTS), SSC 5 mg/kg/day as low dosage (L), 10 mg/kg/day as medium dosage (M), and 20 mg/kg/day as high dosage (H). On the fifth day before antisera injection, mice in groups NTS, L, M, and H were pre-immunized by injection with rabbit IgG and CFA (0.1 mL/mouse), while mice in groups N and SC were injected with normal saline. On the antisera injection day (NTS 0 day), mice in groups NTS, L, M, and H were intravenously injected with 200 μL NTS, and mice in groups N and SC were injected with saline. On the next day of injection (NTS 1st day), mice in groups SC, L, M, and H were administered saikosaponin C once daily for 14 days.

Urine was collected at four intervals, designated baseline (the fourth day before pre-immunization), pre-immune 3rd day (the third day after pre-immunization), NTS 5th day (the fifth day after NTS injection), and NTS 12th day (the twelfth day after NTS injection). Mouse urine was collected using rodent metabolic cages (Tokiwa Chemical Industries Co. Ltd.,Tokyo, Japan); the mice were housed individually with free water access for 12 h at room temperature.

On the fifteenth day after injection, all mice were sacrificed under pentobarbital anesthesia (50 mg/kg body weight). All harvested urine, kidney tissues, and blood samples were stored at -80˚C before analysis [32, 38].

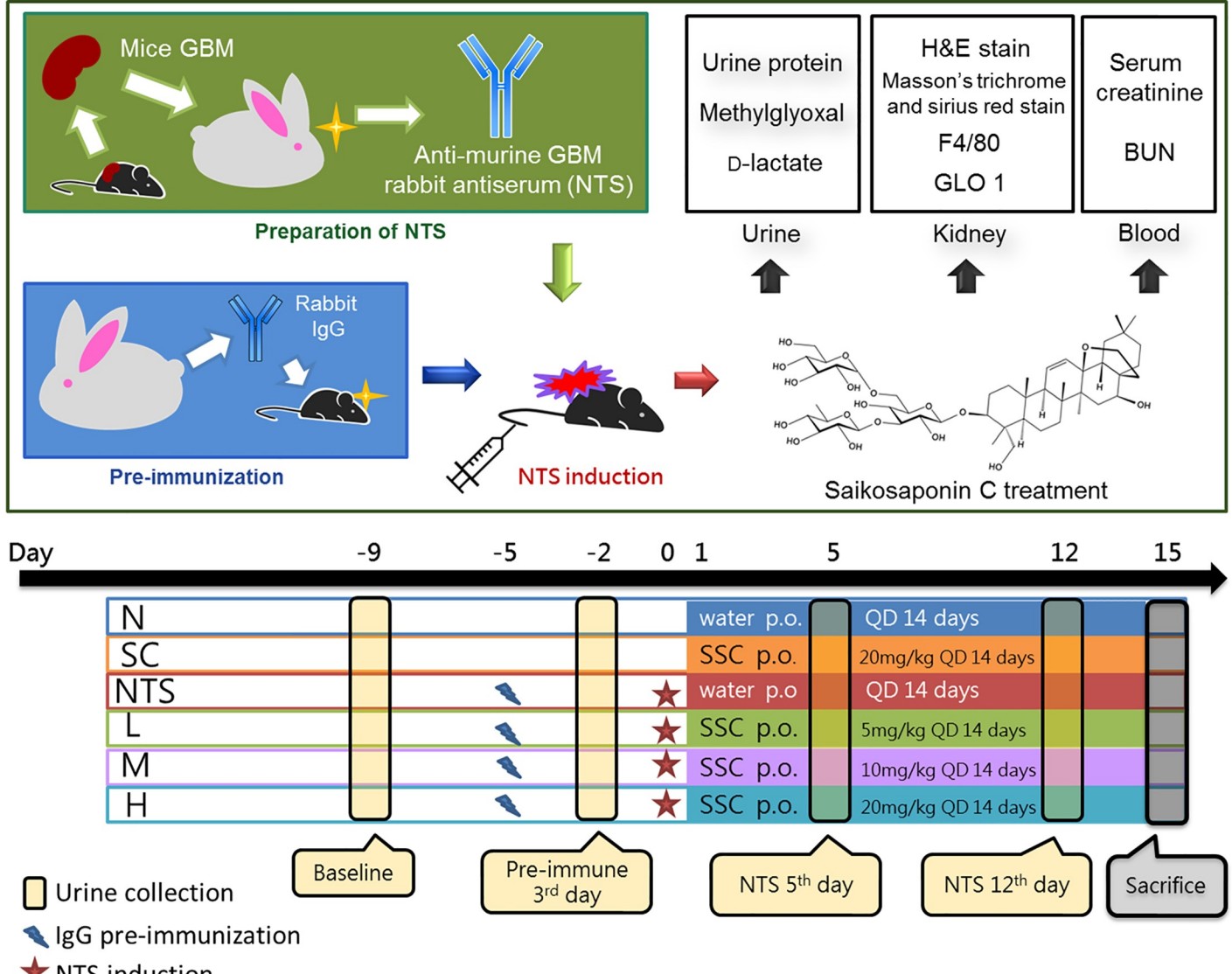

**Fig 1. Schedule of the accelerated nephrotoxic serum nephritis model.** Mice were divided into six groups (*n* = 6): normal (N), NTS control (NTS) saikosaponin C (SSC) control (SC), SSC 5 (L), 10 (M), 20 (H) mg/kg/day. Mice in groups NTS, L, M, and H were pre-immunized by injection with rabbit IgG then were induced with 200 μL NTS intravenous injection, while mice in groups N and SC were injected with normal saline. On the next day, mice in groups SC, L, M, and H were administered SCC once daily for 14 days. The urine was collected at four intervals for analysis of urine protein, methylglyoxal and D-lactate. The serum was collected for biochemical assays as serum creatinine and BUN. The kidneys were collected for histological examination (hematoxylin and eosin, Masson's trichrome, and sirius red stain), immunoblotting analysis of glyoxalase 1 (GLO 1), and immunohistochemistry analysis of F4/80.

### Biochemical assays

All biochemical parameters were measured for each of the 30 mice. Blood urea nitrogen (BUN) and serum creatinine (SCr) were measured by the Animal Experimental Center of Taipei Medical University. Urinary protein was determined with the BCA protein assay according to the manufacturer's instructions. Urinary creatinine was measured using HPLC-UV analysis by modifying a previously described method [42].

## Histological examination

One kidney of each mouse was embedded in paraffin and sliced into 4–5 μm sections [38]. The sections were stained with hematoxylin and eosin (H&E) (Sigma-Aldrich. Inc., Saint Louis, USA), Masson's trichrome stain (Sigma-Aldrich. Inc., Saint Louis, USA), and picro sirius red stain (Cis-Biotechnology Co., Ltd., Taichung, Taiwan) according to the manufacturer's instructions.

## Histological score

Glomerular histological score (GHS): the severity of the glomerular lesions were scored on a scale of 0 (absent) to 5 (extensive change) in three categories (cell proliferation, thrombus, and crescent formation) as previously reported [13]. Tubulointerstitial histological score (THIS): the severity of tubulointerstitial lesions were scored on scales of 0 (absent) to 3 (extensive change) in the two categories (interstitial cells infiltration, and interstitial fibrosis), and a scale of 0 (absent) to 5 (extremely severe degeneration with massive necrosis) in the category in tubular atrophy. The total histological score (THS) was counted with the sum of GHS and THIS.

## Immunohistochemistry analysis

One kidney of each mouse was embedded in paraffin and sliced into 4–5 μm sections. Antibodies against F4/80 (GTX14321; GeneTex, Irvine, USA), and goat anti-rat IgG (H+L) HRP conjugate (112-035-062; Jackson ImmunoResearch Inc., PA, USA) were used at dilutions of 1:100 and 1:2000, respectively. Signals corresponding to the spots for F4/80 were measured by the Dako Liquid DAB+ substrate chromogen system (K346811; Agilent Technologies, Inc., CA, USA).

## Semi-quantitative analysis of fibrosis and macrophage infiltration

Renal fibrosis was assessed in the Masson's trichrome and sirius red stain, and macrophage infiltration was assessed in immunohistochemistry stain. The stained sections were determined the percentage area positive for aniline blue, red, and brown in ten non-overlapping fields of view for each mouse using ImageJ (National Institutes of Health, MD, USA), respectively [39].

## ELISA analysis

Mouse renal tumor necrosis factor α (TNF-α) level was measured using uncoated ELISA kit (Invitrogen, Carlsbad, CA, USA). All reagent preparations and experimental procedures were followed the product information sheets.

## Determination of urinary level of methylglyoxal

Measurement of methylglyoxal was performed using HPLC coupled with fluorescence methods as previously reported [43]. Briefly, a mixture of urine sample, ammonium chloride and 5,6-diamino-2,4-hydroxypyrimidine sulfate was derivatized at 60˚C for 30 min, followed by centrifugation (12000 rpm at 4˚C for 15 min) and filtration (0.2 μm). Each sample was analyzed by HPLC with an ODS column (Biosil, 250 × 4.6 mm i.d.; 5 μm particle size; Biosil Chemical Co. Ltd, Taipei, Taiwan) at 33˚C. The mobile phase comprised 0.01 M citric acid and acetonitrile (97:3, *v/v*; pH adjusted to 5.7). The analysis was operated at excitation and emission wavelengths of 330 and 500 nm, respectively, and the flow rate was 0.7 mL/min.

### Determination of urinary level of D-lactate

A well-established column-switching system [40, 44] was used to measure urinary D-lactate. Briefly, a mixture of 20 μL urine sample, 10 μL propionic acid (internal standard) and 170 μL acetonitrile was centrifuged (1000 *g* at 4˚C for 10 min). Then, 100 μL of the supernatant was derivatized with 50 μL each of 280 mM 2,2'-dipyridyl disulfate and triphenylphosphine, and 100 μL 8 mM 4-nitro-7-piperazino-2,1,3-benzoxadiazole at 30˚C for 3 hours, followed by elution into the MonoSpin™ SCX cartridge (GL Science Inc., Tokyo, Japan) [45]. The eluent was then injected into the column-switching system. For the first dimension, total lactate separation was performed on an ODS column (250 × 4.6 mm i.d.; 5 μm particle size; Biosil Chemical Co. Ltd) at 30˚C. The mobile phase consisted of acetonitrile, methanol, and water (12:20:68, *v/v*), and the flow rate was 0.7 and 0.9 mL/min at 0–35 min and 35.1–60 min, respectively. For the second dimension, chiral lactate were separated with a Chiralpak AD-RH column (150 × 4.6 mm i.d.; 5 μm particle size; Daicel Co., Osaka, Japan) at room temperature (25˚C). The mobile phase was acetonitrile and water (60:40, *v/v*) and the flow rate was 0.3 mL/min. Excitation and emission wavelengths were 491 and 547 nm, respectively.

### Immunoblotting analysis

Renal homogenate protein samples (10 mg) for each group were separated on 12% sodium dodecyl sulfate-polyacrylamide (SDS-PAGE) gels with an SDS-PAGE system [46] and subsequently transferred to nitrocellulose membranes. Antibodies against glyoxalase 1 (GLO1; GTX105792; 19 GeneTex, Irvine, USA), β-actin (20536-1-AP; Proteintech, Rosemont, USA), and goat anti-rabbit IgG (H+L) HRP conjugate (SA00001-2; Proteintech, Rosemont, USA) were used at dilutions of 1:2000, 1:3000, and 1:4000, respectively. Signals corresponding to the bands for GLO1 and β-actin were measured by the TOOL Sensitive ECL kit (New Taipei City, Taiwan). The intensities of the bands were quantitated using ImageJ software (National Institute of Health, Bethesda, USA), and the levels of GLO1 were normalized to β-actin.

### Statistical analysis

Results are expressed as means ± standard deviation. The difference in means was determined using the Kruskal-Wallis test; *p*-values less than 0.05 indicate statistical significance. All data analysis was performed using Statistical Product and Service Solutions (SPSS) for Windows, 17th version (IBM, New York, USA).

## Results

### Biochemical assays

**Urinary protein.**   The 24-h urinary protein content was estimated from the protein measurements obtained from 12-h urine collections (Fig 2). There was no significant difference between the six groups at baseline. Urinary protein was significantly higher in the NTS group than group N on the NTS 5th day (37.94 ± 4.91 vs. 13.67 ± 0.77 mg/day, *p*<0.05) and the NTS 12th day (34.40 ± 6.85 vs. 12.65 ± 2.30 mg/day, *p*<0.05). In the three SSC treatment groups, only group M had a significantly lower urinary protein than the NTS group on the NTS 12th day (17.33 ± 4.79 vs. 34.40 ± 6.85 mg/day, *p*<0.05).

**BUN and serum creatinine.**   There was no significant difference in the BUN level between the six groups on the 15th day after NTS injection, although an increasing trend was observed in the NTS group compared to group N. Also, there was no significant difference in serum creatinine between the six groups, but an increasing trend was again observed in the NTS group

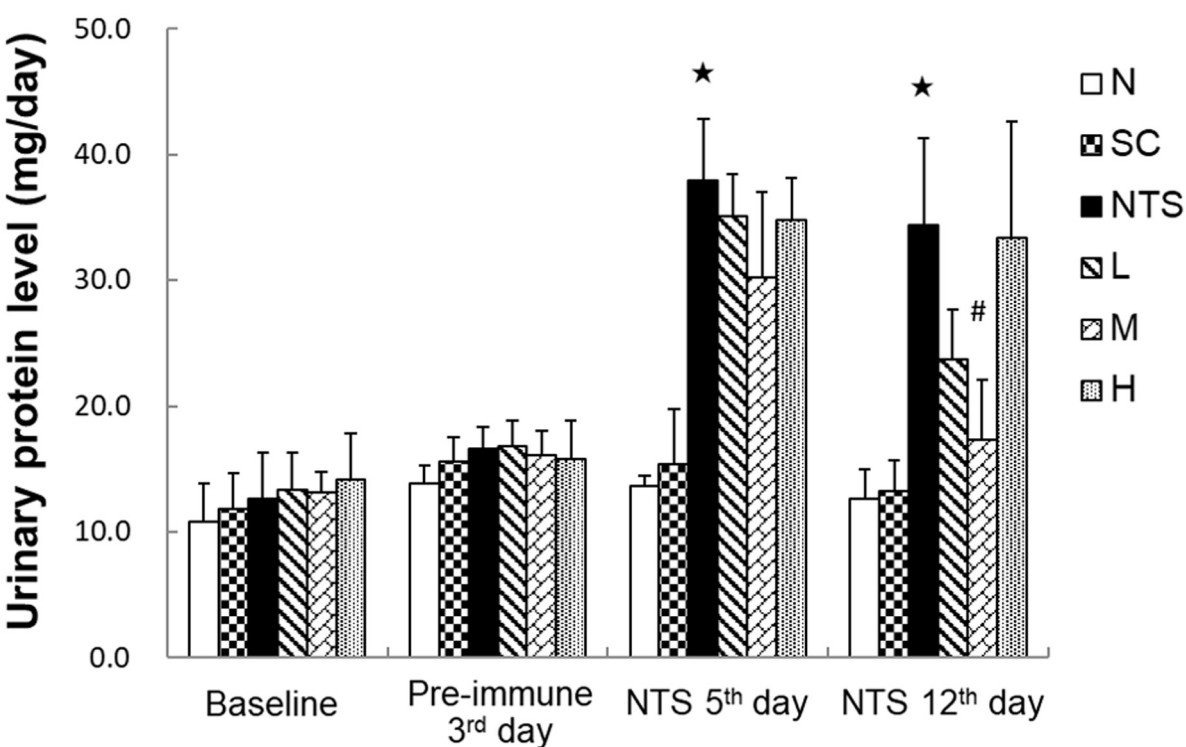

**Fig 2. Urinary protein levels of each experimental group.** N: normal group, SC: saikosaponin C control group, NTS: nephrotoxic serum control group, L: SSC 5 mg/kg/day treatment group, M: SSC 10 mg/kg/day treatment group, H: SSC 20 mg/kg/day treatment group. ★$p < 0.05$ compared to N group, #$p < 0.05$ compared to NTS control group.

compared to group N, while a decreasing trend was observed in the three SSC treatment groups compared to the NTS group (Table 1).

## Histological examination

**Hematoxylin and eosin stain.** Pathological observation of the glomerulus following H&E staining revealed monocyte infiltration, hypertrophy, fission, and even formation of crescents in the NTS-induced groups. However, the glomeruli lesions were less severe in group M (Fig 3E).

Histological scores are shown in Fig 3G. The injury score on the fifteenth day after NTS injection was significantly higher in the NTS group compared to group N, and only group M had a significantly lower injury score than the NTS group (N [0.33 ± 0.37] vs. NTS [7.92 ± 1.37]★ vs. M [3.50 ± 1.14]#; ★$p < 0.05$ compared to N group, #$p < 0.05$ compared to NTS group).

**Table 1. BUN and serum creatinine (SCr) levels of each experimental group.**

| Biochemical test | N | SC | NTS | L | M | H |
|---|---|---|---|---|---|---|
| BUN (mg/dL) | 25.00 ± 8.71 | 27.10 ± 7.01 | 42.60★ ± 17.38 | 41.60 ± 10.56 | 40.00 ± 12.79 | 44.25 ± 16.99 |
| SCr (mg/dL) | 0.21 ± 0.06 | 0.23 ± 0.07 | 0.32 ± 0.11 | 0.23 ± 0.04 | 0.25 ± 0.05 | 0.22 ± 0.04 |

N: normal group, SC: saikosaponin C control group, NTS: nephrotoxic serum control group, L: SSC 5 mg/kg/day as low dosage group, M: SSC 10 mg/kg/day as medium dosage group, H: SSC 20 mg/kg/day as high dosage group.

★$p < 0.05$ compared to N group.

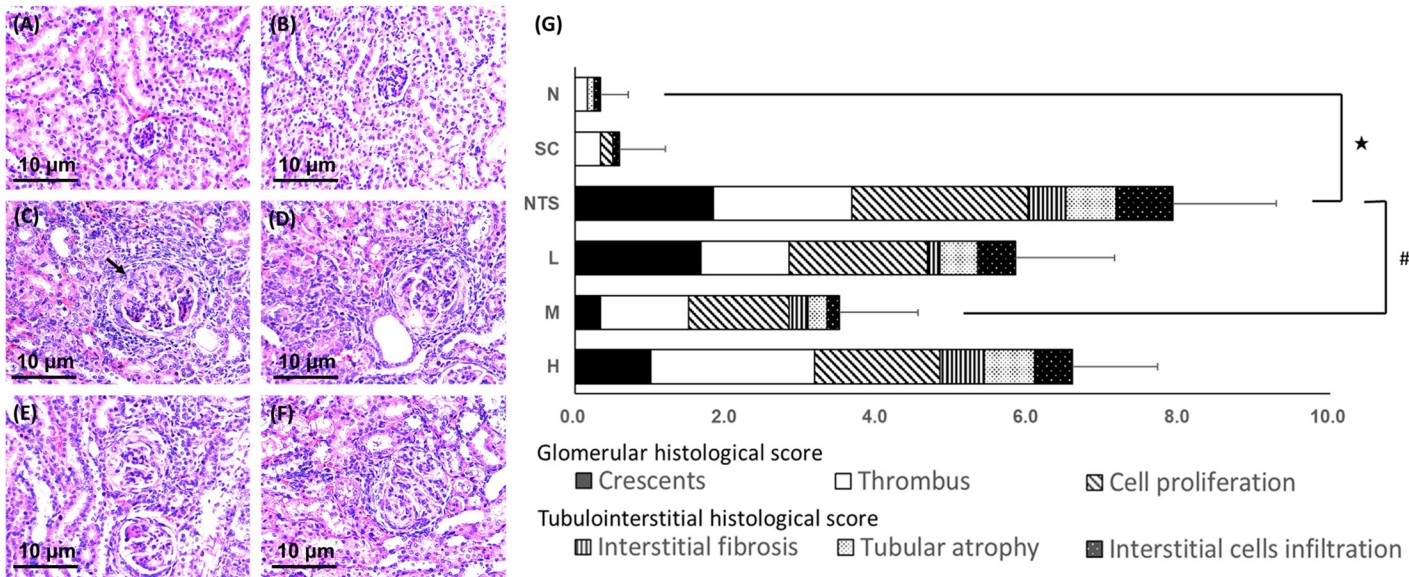

**Fig 3. Hematoxylin and eosin stained renal tissue from different experimental groups (×200 magnification).** (A) Normal, (B) SSC control, (C) NTS control (arrow: the crescent), (D) NTS + SSC 5 mg/kg, (E) NTS + SSC 10 mg/kg, and (F) NTS + SSC 20 mg/kg. (G) Histological scores for the experimental groups. ★$p<0.05$ compared to N group, #$p<0.05$ compared to NTS control group.

**Masson's trichrome stain.** Collagen deposition surrounding the glomerulus and renal interstitium was observed in the NTS group using Masson's trichrome staining (Fig 4C). Semi-quantitative analysis revealed a significantly percentage larger area of collagen deposition in the NTS group compared to group N, and a significantly lower percentage area of collagen deposition in group M than in the NTS group (N [1.02 ± 0.94%] vs. NTS [11.16 ± 3.02%]★ vs. M [4.67 ± 1.31%]#; ★$p<0.05$ compared to N group, #$p<0.05$ compared to NTS group; Fig 4G).

**Sirius red stain.** Collagen deposition was also observed in the NTS group using sirius red staining (Fig 5C). Semi-quantitative analysis revealed a significantly larger area of collagen deposition in the NTS group compared to group N, and a lower area of collagen deposition in group M than in the NTS group (N [1.38 ± 0.36%] vs. NTS [10.13 ± 4.16%]★ vs. M [5.25 ± 2.38%]#; ★$p<0.05$ compared to N group, #$p<0.05$ compared to NTS group; Fig 5G). Based on these experimental data, the M dosage (10 mg/kg/day) was selected as the optimal treatment dose for NTS nephritis from the three SSC groups.

**Immunohistochemistry stain.** The infiltration of macrophage was observed in the NTS group using immunohistochemistry staining (Fig 6C). Semi-quantitative analysis revealed a significantly larger area of macrophage infiltration in the NTS group compared to group N, and a less area of macrophage infiltration in group M than in the NTS group (N [0.068 ± 0.036%] vs. NTS [1.534 ± 0.523%]★ vs. M [0.624 ± 0.702%]#; ★$p<0.05$ compared to N group, #$p<0.05$ compared to NTS group; Fig 6G).

**Determination of renal level of TNF-α.** Renal TNF-α measurements are shown in Fig 7. The level of TNF-α in renal tissue was significantly higher in the NTS group than the normal group, and only group M was observed significantly lower levels of TNF-α compared to the NTS group (N [0.12 ± 0.06] vs. NTS [0.30 ± 0.07]★ vs. M [0.16 ± 0.05]#; ★$p<0.05$ compared to N group, #$p<0.05$ compared to NTS group).

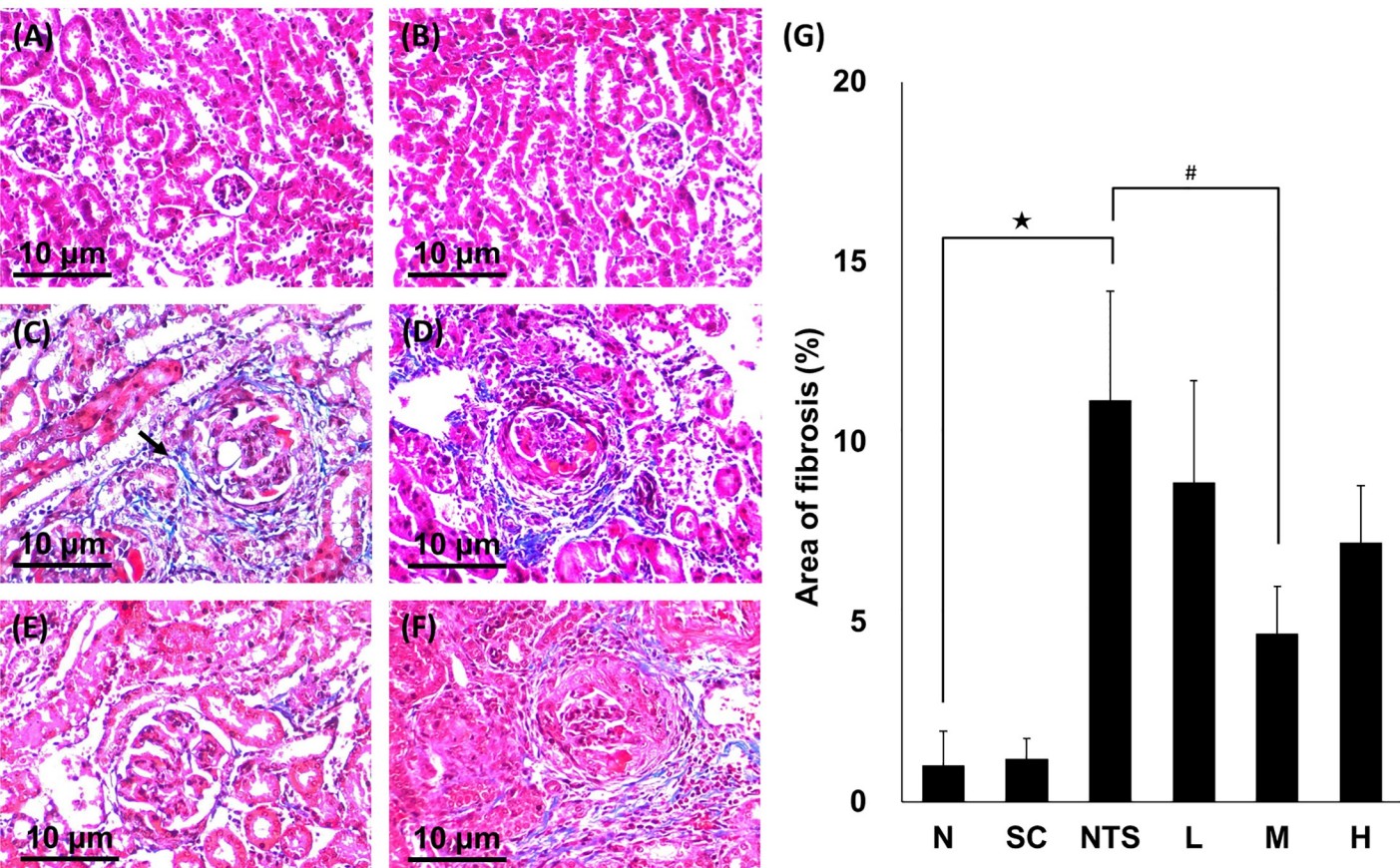

**Fig 4. Masson's trichrome-stained renal tissues from different experimental groups (×200 magnification).** (A) Normal, (B) SSC control, (C) NTS control (arrow, the part of fibrosis), (D) NTS + SSC 5 mg/kg, (E) NTS + SSC 10 mg/kg, and (F) NTS + SSC 20 mg/kg. (G) Quantification of the area of fibrosis for each experimental group. ★$p<0.05$ compared to normal group, #$p<0.05$ compared to NTS control group.

## Determination of urinary level of methylglyoxal

Urinary MGO measurements are shown in Fig 8. There was no significant difference between the six groups at baseline. On the NTS 5th day, urinary MGO was significantly higher in the NTS group than group N, while only group M had a significantly lower level compared to the NTS group (N [8.42 ± 1.74 μg/mg] vs. NTS [30.12 ± 4.10 μg/mg]★ vs. M [17.13 ± 4.60 μg/mg]#; ★$p<0.05$ compared to N group, #$p<0.05$ compared to NTS group). On the NTS 12th day, the level of urinary MGO was still significantly higher in the NTS group than the normal group, and only group M had a significantly lower level compared to the NTS group (N [8.96 ± 2.69 μg/mg] vs. NTS [24.71 ± 3.46 μg/mg]★ vs. M [16.72 ± 2.36 μg/mg]#; ★$p<0.05$ compared to N group, #$p<0.05$ compared to NTS group).

## Determination of urinary level of D-lactate

Urinary D-lactate levels are shown in Fig 9. There was no significant difference between the six groups at baseline. On the NTS 5th day, urinary D-lactate was significantly higher in the NTS group than group N, while only group M had a significantly reduced level compared to the NTS group (N [0.15 ± 0.04 μmol/mg] vs. NTS [0.37 ± 0.05 μmol/mg]★ vs. M [0.26 ± 0.03 μmol/mg]#; ★$p<0.05$ compared to N group, #$p<0.05$ compared to NTS group). On the NTS 12th day, the level of urinary D-lactate was still significantly higher in the NTS group than

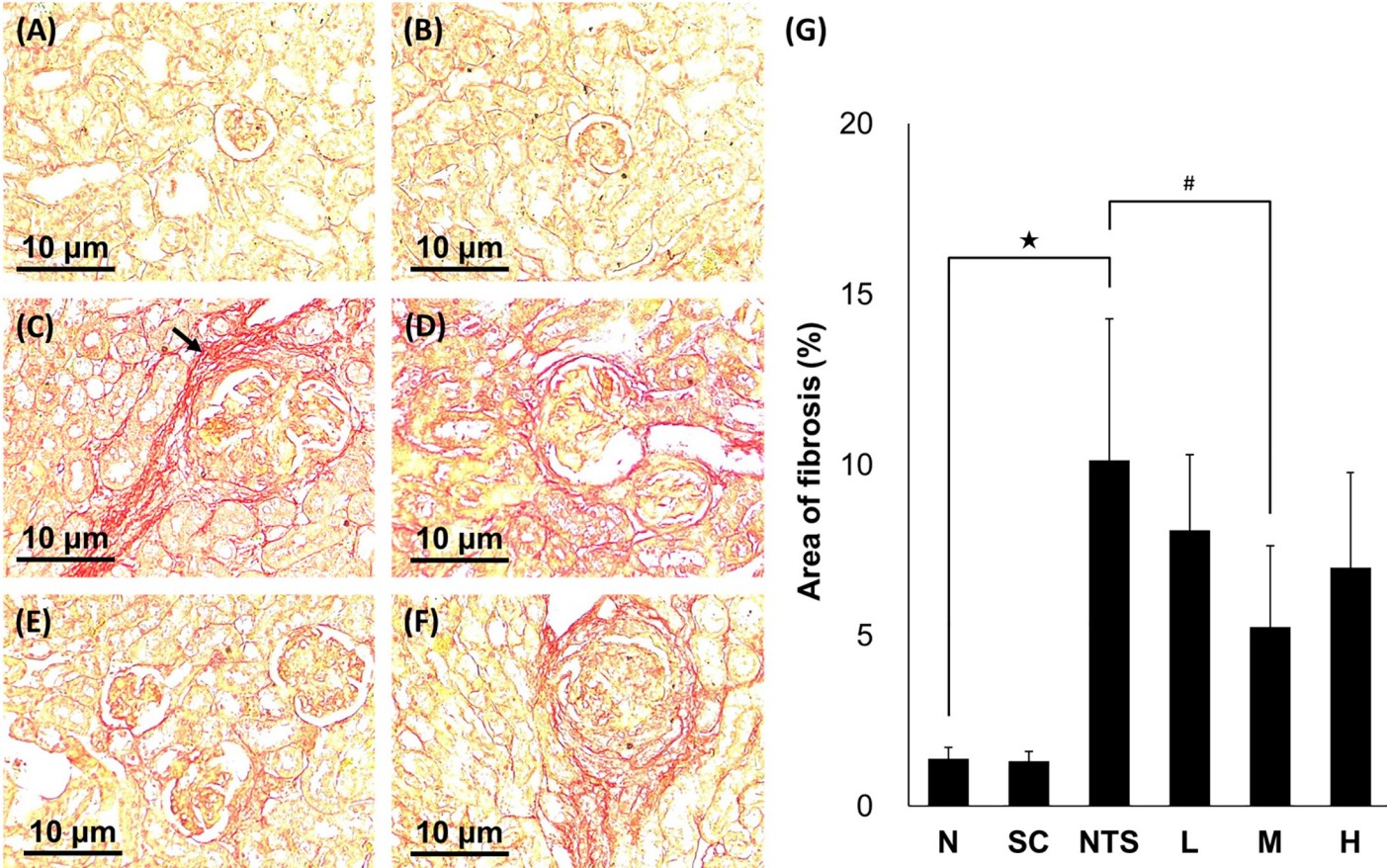

**Fig 5. Sirius red-stained renal tissues from different experimental groups (×200 magnification).** (A) Normal, (B) SSC control, (C) NTS control (arrow, the part of fibrosis), (D) NTS + SSC 5 mg/kg, (E) NTS + SSC 10 mg/kg, and (F) NTS + SSC 20 mg/kg. (G) Quantification of the area of fibrosis for each experimental group. ★$p < 0.05$ compared to normal group, #$p < 0.05$ compared to NTS control group.

the normal group, and only group M had a significantly lower level compared to the NTS group (N [0.12 ± 0.03 μmol/mg] vs. NTS [0.31 ± 0.04 μmol/mg]★ vs. M [0.23 ± 0.02 μmol/mg]#; ★$p < 0.05$ compared to N group, #$p < 0.05$ compared to NTS group).

## Immunoblotting analysis

GLO1 expression in renal tissue was significantly higher in the NTS group than the normal group (4.90 ± 2.12, $p < 0.05$ compared to N group), and only group M expressed significantly lower levels of GLO1 compared to the NTS group (1.50 ± 0.47, $p < 0.05$ compared to NTS group; Fig 10B).

## Discussion

In this study, glomerulus damage was successfully induced via injection of nephrotoxic serum, and the symptom of proteinuria was similar to a previous study [11]. The glomerular filtration system can be roughly divided (moving from the inside outwards) into endothelial cells, basement membrane and podocytes. Since cations pass through the basement membrane at a slower rate and form an electrical potential gradient, negatively charged proteins are repelled by the filtration barrier and remain in the blood [47]. Damage to podocytes disrupts the filtration barrier and leakage of protein may be detectable in the urine [48]. Feng *et al.* reported that

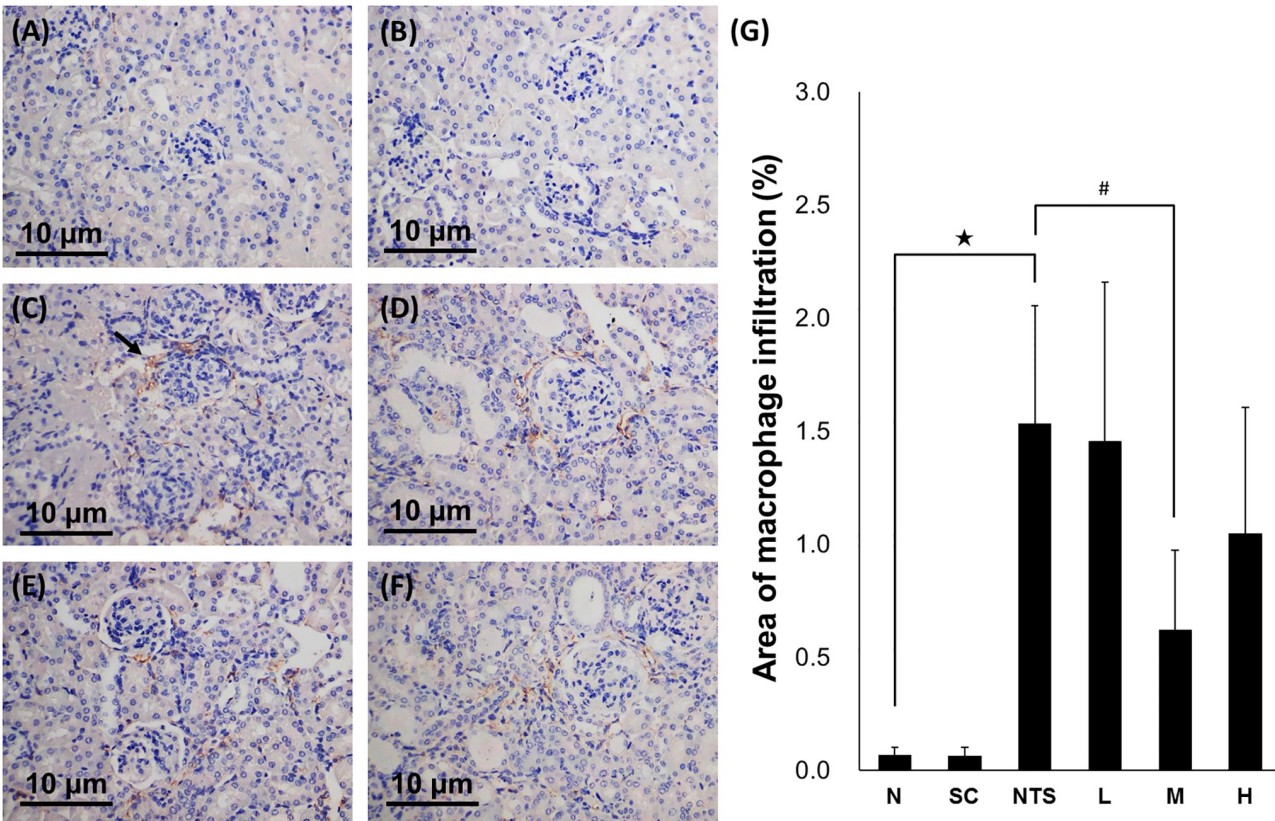

**Fig 6. Immunohistochemistry stained renal tissues for F4/80 from different experimental groups (×200 magnification).** (A) Normal, (B) SSC control, (C) NTS control (arrow, the part of macrophage infiltration), (D) NTS + SSC 5 mg/kg, (E) NTS + SSC 10 mg/kg, and (F) NTS + SSC 20 mg/kg. (G) Quantification of the area of fibrosis for each experimental group. ★$p < 0.05$ compared to normal group, #$p < 0.05$ compared to NTS control group.

the severity of NTS proteinuria is positively related to podocyte injury. In the electronic microscopic examination, the podocytes showed extensive flattening and fusion in mice those occurred proteinuria [49]. Therefore, the proteinuria observed in the present study may indicated the presence of injured podocytes and glomerular dysfunction.

Although the serum creatinine and BUN levels in the NTS group were not significantly different from the other experimental groups, the upward trend was similar to that observed in previous studies [11, 13]. Clinically, a decrease in BUN and serum creatinine occur in late-stage nephropathy, whereas proteinuria is a better indicator of early-stage nephropathy. Our data suggest that severe fibrosis had not occurred in all glomeruli, and that some uninvaded nephrons maintained physiological function.

According the H&E staining, some hypertrophic glomeruli, monocyte infiltration, and even crescents were observed, and the interstitial cell infiltration, and fibrosis were also occurred under NTS injury. Besides, significant collagen deposition in the renal interstitium and surrounding the glomerulus was observed in the NTS group via Masson's trichrome and sirius red staining. These findings indicate that the autoimmune nephritis model may accelerate the development of fibrosis in some nephrons.

The trend of renal TNF-α, a biomarker of inflammation, was consistent to the histological finding. The NTS induced renal damage in histological observation is related to inflammation.

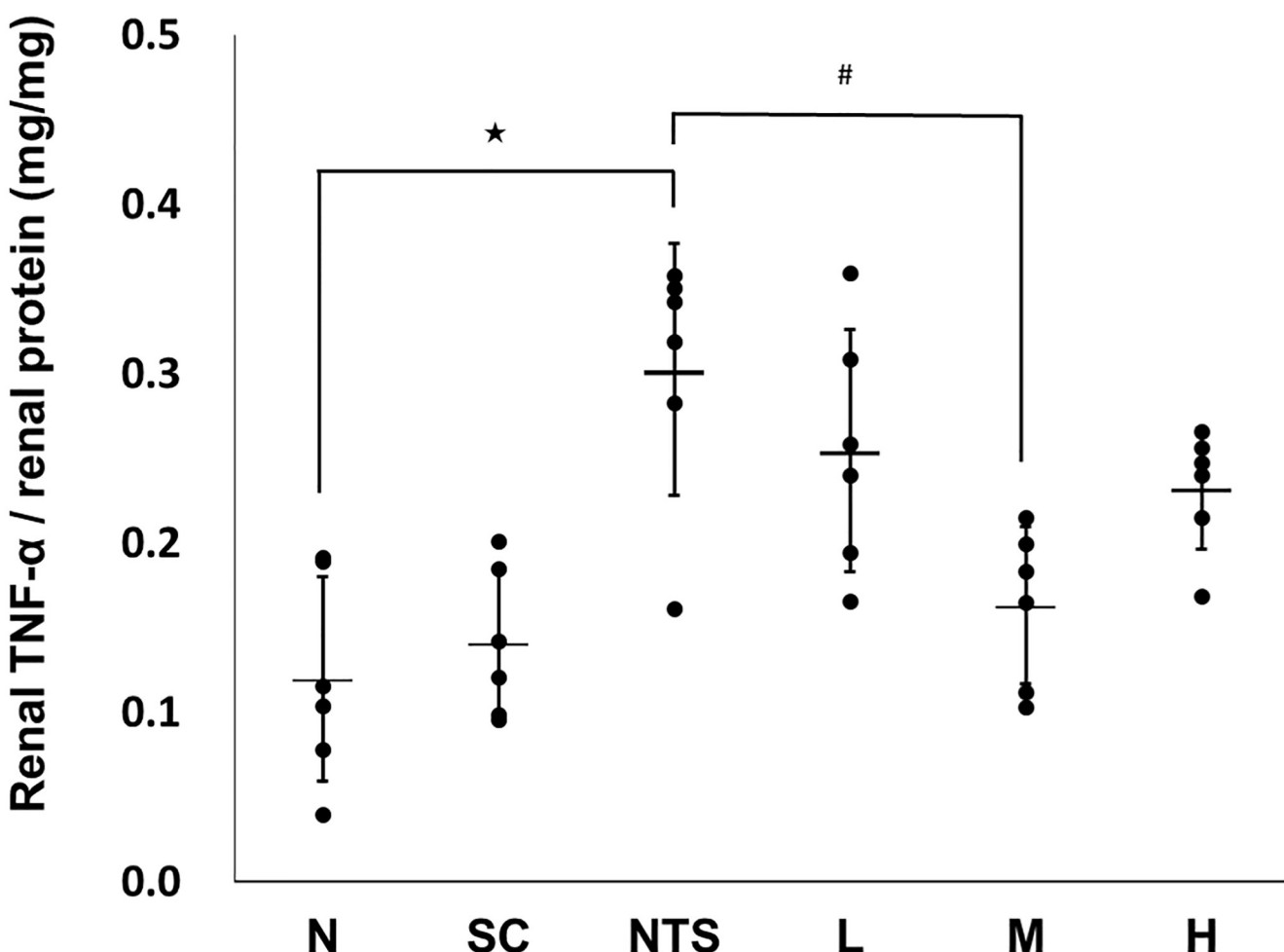

**Fig 7. Renal TNF-α levels in the different experimental groups.** N: normal group, SC: saikosaponin C control group, NTS: nephrotoxic serum control group, L: SSC 5 mg/kg/day treatment group, M: SSC 10 mg/kg/day treatment group, H: SSC 20 mg/kg/day treatment group. ★$p<0.05$ compared to normal group, #$p<0.05$ compared to NTS control group.

The similar trend was observed in renal level of F4/80, a biomarker of macrophage. It revealed that the level of macrophage infiltration induced by NTS injection is positively correlated with nephritis severity as previous reports noted [50]. Besides, the depletion of macrophage may improve disease in several animal models of autoimmune related nephritis [50, 51]. In addition, it was observed that SSC may partly ameliorate the disease progression and decrease the level of macrophage infiltration, although it did not work in all treatment groups.

The NTS nephritis model has been partially explained. Firstly, γδ T cells enter the kidney and release interleukin-17, which attracts neutrophils to migrate into the glomerulus, causing damage [52]. Subsequently, T helper-17 cells enter the renal interstitium and continuously promote neutrophil infiltration. Dendritic cells then activate T helper-1 cells and cause macrophage infiltration, leading to subsequent damage [53]. In this study, we observed that SSC may ameliorate NTS nephritis in mice as reported by Chen *et al.*, and the efficacy is probably due to the steroid-like structure of SSC [13]. It was previously noted that the phytosterols with triterpene structure may alleviate inflammation via activation of the glucocorticoid receptor (GR) [54]. According to a recent study of aristolochic acid nephropathy, prednisolone may alleviate renal damage and fibrosis and inhibit glycolysis [39]. Moreover, the decreases in MGO, GLO1

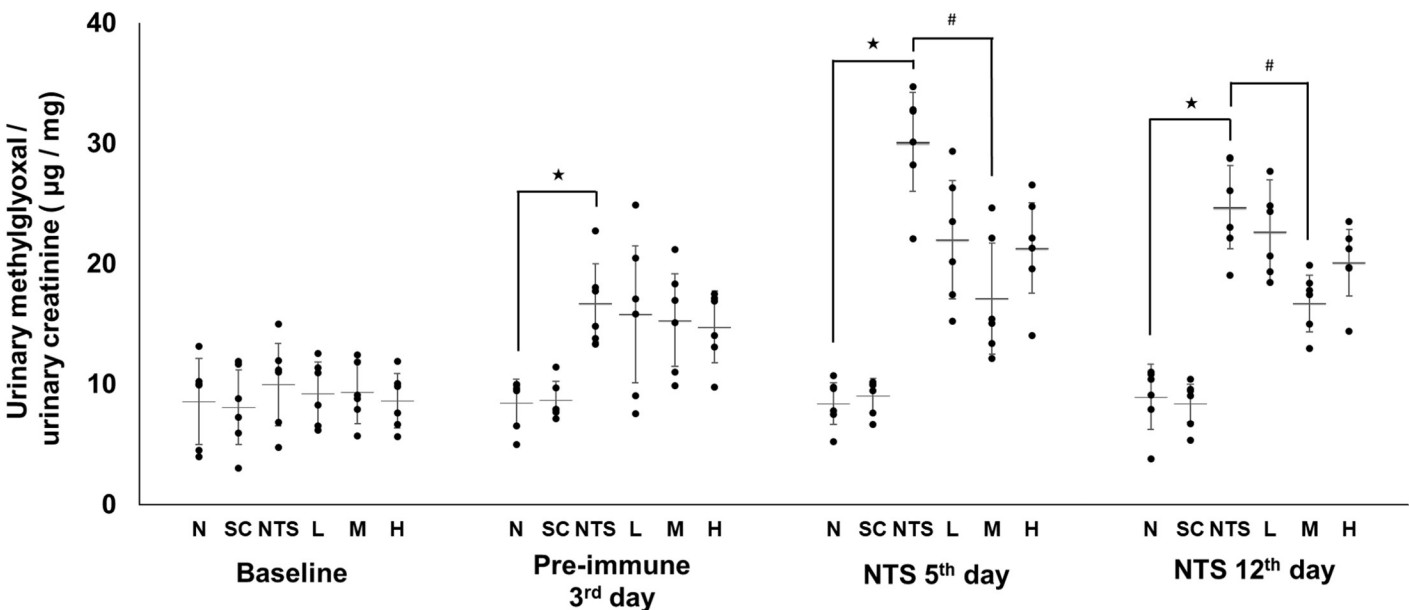

**Fig 8. Urinary methylglyoxal levels in the different experimental groups.** N: normal group, SC: saikosaponin C control group, NTS: nephrotoxic serum control group, L: SSC 5 mg/kg/day treatment group, M: SSC 10 mg/kg/day treatment group, H: SSC 20 mg/kg/day treatment group. ★$p < 0.05$ compared to normal group, #$p < 0.05$ compared to NTS control group.

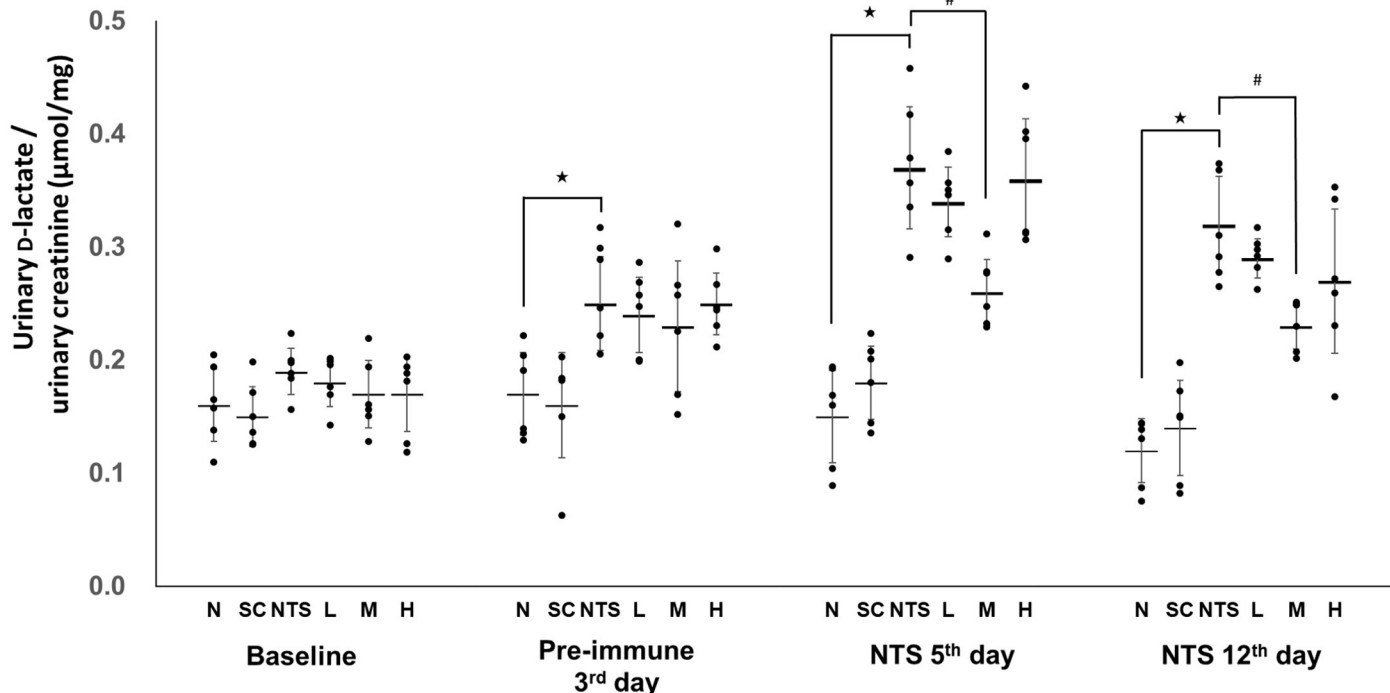

**Fig 9. Urinary D-lactate levels in different experimental groups.** N: normal group, SC: saikosaponin C control group, NTS: nephrotoxic serum control group, L: SSC 5 mg/kg/day treatment group, M: SSC 10 mg/kg/day treatment group, H: SSC 20 mg/kg/day treatment group. ★$p < 0.05$ compared to normal group, #$p < 0.05$ compared to NTS control group.

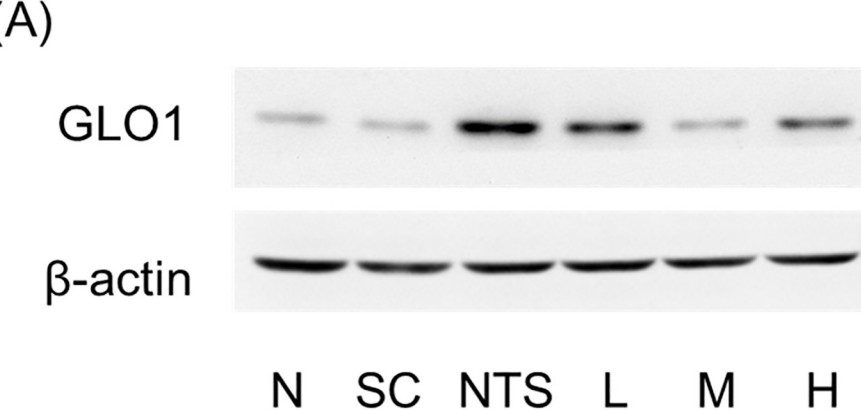

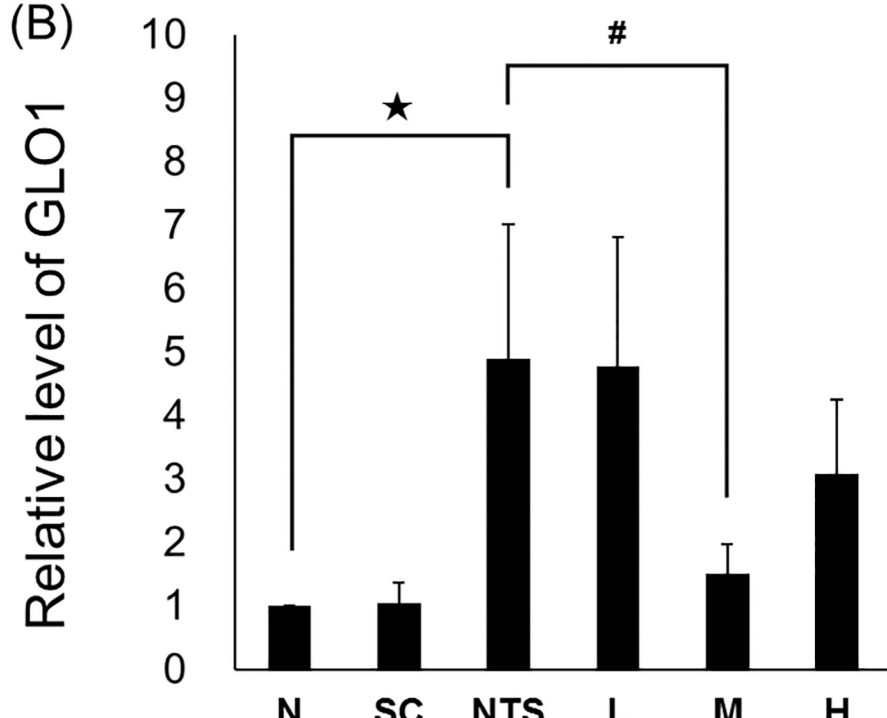

**Fig 10. Immunoblotting analysis of glyoxalase 1 in renal homogenates from different experimental groups on day 15.** N: normal group, SC: saikosaponin C control group, NTS: nephrotoxic serum control group, L: SSC 5 mg/kg/day treatment group, M: SSC 10 mg/kg/day treatment group, H: SSC 20 mg/kg/day treatment group. $\star p < 0.05$ compared to normal group, $^{\#}p < 0.05$ compared to NTS control group.

and D-lactate with prednisolone treatment were consistent with the effects of SSC treatment in NTS nephritis in the present study. Thus, the anti-inflammatory effect of SSC may be related to GR activation and inhibition of glycolysis.

The present study revealed that only the middle-dose (10 mg/kg) SSC group showed a significant improvement effect in the NTS nephritis model, and the low-dose (5 mg/kg) SSC group did not. This indicates that the anti-inflammatory effect of SSC has a dose-related threshold in NTS nephritis. However, it was noted that the ameliorative effect on renal damage

and fibrosis reduced with the high-dose (20 mg/kg) SSC treatment. This might be related to the varied effects of GR such as inducing survival of neutrophils [55] and apoptosis of T or B cells [56]. According to a recent study, inhibition of GR may ameliorate glomerulonephritis and prevent crescent formation [12]. This indicates that activation of the GR is involved in glomerular crescent formation and fibrosis, although the precise mechanism is unclear. In addition, TNF-α is also involved in downregulating the immune response in NTS nephritis, the TNF-α knock out T cell may cause more severe nephritis [57]. A high dose of SSC might disrupt the balance of GR- or TNF-α- regulated systems, and counteract the anti-inflammatory benefits in nephritis. In present study, there was no difference in high dose of SSC treatment group in serum creatinine and BUN examinations. Although the proteinuria in H group was not improved, at least it was not worse compared to NTS group. The study revealed that SSC may lose the protective effect in high dose from NTS nephritis. Thus, administration of high-dose SSC does not further improve the efficacy of treatment in NTS nephritis.

MGO and D-lactate are indicators of renal damage, as previously reported [35, 38, 58]. In the present study, the trends of MGO and D-lactate levels correlated positively with renal damage and were reversed with SSC treatment. Glycolysis is enhanced during renal damage, since inflammation and repair require a high consumption of energy [59]. During glycolysis, glucose is first converted into fructose 1,6-bisphosphate via phosphorylation [60]. Then, aldolase B catalyzes the cleavage of fructose 1,6-bisphosphate into dihydroxyacetone phosphate (DHAP)

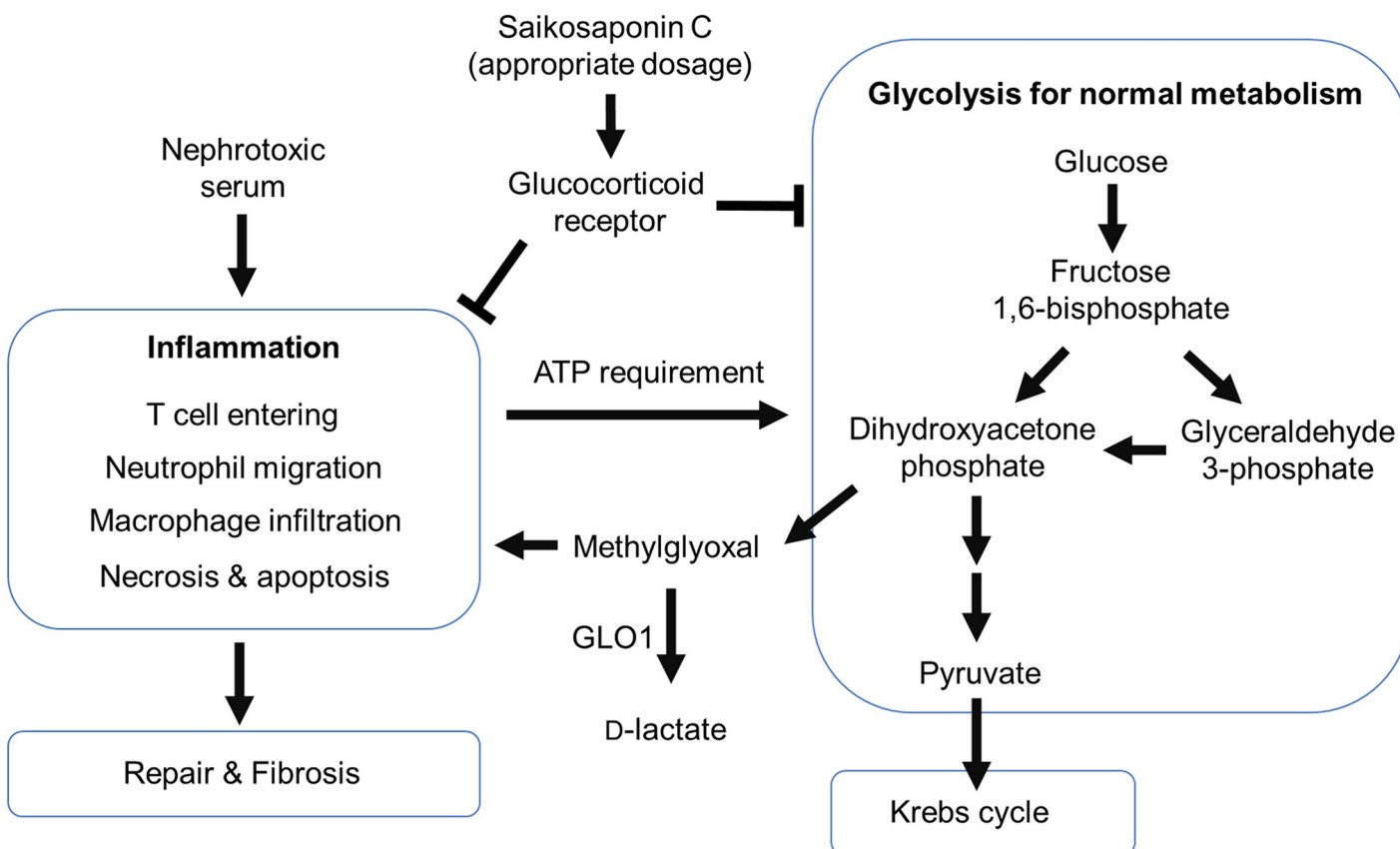

**Fig 11. The relationship between methylglyoxal and D-lactate in saikosaponin C treatment of nephrotoxic serum nephritis.** Nephrotoxic serum nephritis induced inflammation up-regulated glycolysis then cause methylglyoxal and D-lactate increased in the urine. The appropriate dosage of saikosaponin C treatment may improve the immune-reaction and down-regulate methylglyoxal and D-lactate production.

and glyceraldehyde 3-phosphate (GAP). Some DHAP is converted into GAP by triose-phosphate isomerase and proceeds into the glycolysis pathway, while some DHAP is bypassed for production of MGO by methylglyoxal synthase [61]. Therefore, glycolysis not only provides energy, but also produces toxic MGO via the bypass pathway. GLO1, a rate-limiting enzyme, catalyzes the conversion of methylglyoxal- Glutathione hemithioacetal into the thioester S-D-lactoylglutathione, followed by metabolism of glyoxalase 2 (GLO2), producing D-lactate and Glutathione [30]. Thus, GLO1 plays a key role in protection from MGO attack. This explains the similar trends in GLO1 expression in renal tissue and urinary MGO and D-lactate levels in this study.

## Conclusion

In summary, this study demonstrates that the middle dose of SSC (10 mg/kg) may ameliorate renal damage and fibrosis in the accelerated NTS nephritis model. This effect was reflected in the levels of urinary MGO and D-lactate. Fig 11 outlines current understanding of the relationship between MGO and D-lactate in SSC treatment of NTS nephritis. Further investigations on the efficacy and adverse effects of SSC are required before clinical application is possible.

## Supporting information

**S1 Raw images. Raw gel images of Fig 10.**
(PDF)

## Acknowledgments

We appreciate that Prof Shiro Ueda supplied our team the antisera and the suggestion about metabolic cages.

## Author Contributions

**Conceptualization:** Shih-Ming Chen.

**Data curation:** Po-Yeh Lin, Shih-Ming Chen.

**Formal analysis:** Chia-Yu Lin, Shih-Chun Hua, Pei-Yun Tsai, Bi-Li Chen, Chia-En Lin.

**Funding acquisition:** Shih-Ming Chen.

**Methodology:** Jen-Ai Lee, Po-Yeh Lin, Pei-Yun Tsai.

**Supervision:** Jen-Ai Lee, Tzong-Huei Lee, Shih-Ming Chen.

**Writing – original draft:** Chia-Yu Lin, Shih-Ming Chen.

**Writing – review & editing:** Jen-Ai Lee, Tzong-Huei Lee, Shih-Ming Chen.

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
