## [Decision Letter · Decision Letter 0]

28 Apr 2020

PONE-D-20-05825

Utilizing Methylglyoxal and D-lactate in Urine to Evaluate Saikosaponin C Treatment in Mice with Accelerated Nephrotoxic Serum Nephritis

PLOS ONE

Dear Doctor Shih-Ming Chen,

Thank you for submitting your manuscript to PLOS ONE. After careful consideration, we feel that it has merit but does not fully meet PLOS ONE’s publication criteria as it currently stands. Therefore, we invite you to submit a revised version of the manuscript that addresses the points raised during the review process.

After careful reading I think that the paper is of interest. However, some major points have been raised by the reviewers. Please answer point by point all the following issues before reconsidering your manuscript.

We would appreciate receiving your revised manuscript by the end of September 2020. To enhance the reproducibility of your results, we recommend that if applicable you deposit your laboratory protocols in protocols.io, where a protocol can be assigned its own identifier (DOI) such that it can be cited independently in the future. For instructions see: http://journals.plos.org/plosone/s/submission-guidelines#loc-laboratory-protocols

We look forward to receiving your revised manuscript.

Kind regards,

Christos E. Chadjichristos

Academic Editor

PLOS ONE

Journal Requirements:

Additional Editor Comments (if provided):

After careful reading I think that the paper is of interest. However, some major points have been raised by the reviewers. Please answer point by point all the following points before reconsidering your manuscript.

Reviewers' comments:

Reviewer's Responses to Questions

**Comments to the Author**

1. Is the manuscript technically sound, and do the data support the conclusions?

Reviewer #1: Partly

Reviewer #2: Yes

2. Has the statistical analysis been performed appropriately and rigorously? 

Reviewer #1: N/A

Reviewer #2: Yes

3. Have the authors made all data underlying the findings in their manuscript fully available?

Reviewer #1: Yes

Reviewer #2: Yes

4. Is the manuscript presented in an intelligible fashion and written in standard English?

Reviewer #1: Yes

Reviewer #2: Yes

5. Review Comments to the Author

Reviewer #1: Chia-Yu Lin et al are interested on Saikosaponin (SSC) a bioactive element of the medicinal plant Bupleurum falcatum L, largely used in traditional medicine in Asia for the treatment of various diseases. Here they investigate the effect of three SSC doses, on mice with accelerated nephrotoxic serum nephritis (NTS) as well as the relationship between Methylglyoxal (MGO) and D-lactate, two compounds known to be related to renal injury. Indeed, the authors demonstrate that urinary MGO and D-lactate levels correlate with renal damages.

Although the SSC treatment does not completely prevent disease development, they convincingly show the effect of an intermediate dose of SSC in delaying disease progression. However, not all the aspects of the manuscripts are sufficiently developed and some conclusions are not supported by data, or further investigated. In addition, some clarifications are needed, in particular with regards to the relationship between MGO and L-lactate, there is no evidence of any direct link between them.

Nevertheless, the study remains interesting, but the authors should improve their characterization and perform more analysis. Overall, the validation of the presented results with more data would improve the relevance of the manuscript.

Major comments

- It is somehow counterintuitive that only the intermediate, but not the high dose, of SSC is protective. This should be discussed. Potential toxic effect of the high dose should be evaluated in more detail (i.e. biochemical analysis or body weight).

- The authors state that the intermediate dose decreases the inflammatory immune infiltration and inflammation. However, these data are not shown and/or quantified.

- Mechanistic insights

Figure2: Are urine collected at the end-time point? What about proteinuria at day 15 (end of experiment)?

Figure 3-Figure 4

The authors describe NTS induced-damage with and histological score (defined as cell proliferation, thrombus, and crescent formation) and area of fibrosis. It would be interesting to have more information about the histological differences in mice. Are any of the analysed parameters selectively improved by the treatment with SSC? What about tubular or glomerular damage? As I have already pointed out, the authors mention differences in the inflammatory process but this is not support by any data/quantification. Staining for major immune populations should be done.

Figure5-6: Same as in figure 2. Did the authors measure these urinary parameters at end-point (day +15)?

"The appropriate dosage of saikosaponin C treatment may improve the immune-reaction and down-regulate methylglyoxal and D-lactate production"

In the discussion the authors explain the anti-inflammatory effect of SSC may be related to glucocorticoid receptor (GR) activation and inhibition of glycolysis. This is a hasty conclusion that can be true but this is not supported by any data/quantification for anti-inflammatory effect or any link with GR.

Figure7:

The WB blot shown for glyoxalase 1(GLOT1) is not representative of the quantification provided. Glot1 seems more expressed in SC group compared to the N group. The GLOT1 expression looks similar in the M and H grouped. Was GLOT1 expression evaluated in all the treated mice? Additional WB image should be provided.

Minor comment

I would suggest to display individual data points in graphs (dot plot).

Reviewer #2: The paper presented by Lin et al aims to evaluate the effect of different dosages of SSC on mice with accelerated NTS nephritis, and the relationship between urinary MGO and D-lactate during SSC treatment of NTS nephritis.

The paper is simple to read, written almost good English. Data are generally presented clearly, and the analyses well described and reported in the text. I propose some major revisions to make the paper suitable for publication in Plos One.

- The images must absolutely re edited clearly. The colors are too strong and the contrast difficult to understand (masson staining has the same colors than H & E!).

- I suggest the evaluation of fibrosis using Sirius red staining.

- The clear explanation of the histological score is mandatory!

- As correctly reported by the authors, probably the degree of proteinuria reflects a podocyte damage. In order to evaluate it and increase the power of the result presented I suggest the evaluation of podocyte marker (f.i. nephrin, synaptopodin) in immunofluorescence of (better) in PCR.

6. PLOS authors have the option to publish the peer review history of their article (what does this mean?). If published, this will include your full peer review and any attached files.

Reviewer #1: No

Reviewer #2: No

---

## [Author Response · Author response to Decision Letter 0]

12 Jun 2020

Reviewer #1:

Response to comment 1: 

We observed that there is no difference in SSC high dose group in serum creatinine and BUN examinations, but a high level of proteinuria in H group. It reveals that high dose of SSC may lose the protective effect from NTS nephritis due to immune-related influence. At least high dose of SSC did not perform worse than NTS injury group in present examinations. The potential toxic effect might be minor in present study.

Response to comment 2: 

We have completed the ELISA analysis of TNF-α, as a quantitation of inflammation. Besides, we have added detailed items of histological score such as cells infiltration for inflammation. 

Response to comment 3-1: 

We collected urine on 4 points (day-9, day-2, day5, and day12 of NTS injection), according to previous study, the severe proteinuria occurred on day 10 to 12. 

(Chen SM et al., Induction of nephrotoxic serum nephritis in inbred mice and suppressive effect of colchicine on the development of this nephritis. Pharmacol Res. 2002;45(4): 319-324. https://doi.org/10.1006/phrs.2002.0948 PMID: 12030796)

(Chen SM et al., Effects of Bupleurum scorzoneraefolium, Bupleurum falcatum, and saponins on nephrotoxic serum nephritis in mice. J. Ethnopharmacol. 2008;116(3): 397-402. https://doi.org/10.1016/j.jep.2007.11.026 PMID: 18262740)

Furthermore, urine collection is a hard loading for nephritis mice, in order to reduce mortality rate, we did not collect urine on day 15 (end of experiment).

Response to comment 3-2: 

We adding the detailed items of histological score, and revised the figure 3 and figure 4 as reviewer suggestion. And we have completed a quantitation of inflammation with TNF-α ELISA analysis as previous description.

Response to comment 3-3:

We did not collect the urine on day 15, and the reasons are described previously.

We have quantitated TNF-alpha as inflammation barker, and anti-inflammation effect of SSC was confirmed. 

Response to comment 3-4:

We evaluated all mice, and picked one sample of each group for the WB image in figure 7. To reduce the operation error for final displayed image, we have added a new WB image with loading pooled samples of each group. The following is the new image of pooled sample.

Response to Minor comment:

Thank you for your suggestion, we have revised the figure of proteinuria, renal TNF-alpha, urinary methylglyoxal, and urinary D-lactate into dot plot as the following.

Reviewer #2:

Response to comment 1: 

Thank you for your suggestion. We have taken new images of H&E and masson trichrome stained sections with brighter aperture, it might be clearer.

Response to comment 2: 

We have completed the evaluation of sirius red staining as reviewer suggestion, the trend of renal fibrosis is similar to masson staining.

Response to comment 3: 

We adding the details of histological score, and revised the figure 3 as reviewer suggestion.

Response to comment 4: 

We are sorry for that our samples are not enough for immunofluorescence evaluation, but we provide additional reference to support that the relationship between the proteinuria severity and podocytes injury in the NTS nephritis model.

(Lin et al., Decay-accelerating factor confers protection against complement-mediated podocyte injury in acute nephrotoxic nephritis. Lab Invest. 2002;82(5):563‐569. https://doi:10.1038/labinvest.3780451 PMID: 12003997)

Besides, we revised the discussion for podocytes as the following.

---

## [Decision Letter · Decision Letter 1]

7 Aug 2020

PONE-D-20-05825R1

Utilizing Methylglyoxal and D-lactate in Urine to Evaluate Saikosaponin C Treatment in Mice with Accelerated Nephrotoxic Serum Nephritis

PLOS ONE

Dear Dr. Shih-Ming Chen,

Thank you for submitting your manuscript to PLOS ONE. After careful consideration, we feel that it has merit but does not fully meet PLOS ONE’s publication criteria as it currently stands. Therefore, we invite you to submit a revised version of the manuscript that addresses the points raised during the review process.

Indeed, after careful reading and based also to the reviewer's suggestions, I think that before publication two issues have to be addressed. Please provide:

1/  immunostainings for inflammatory cells, and

2/  scale bars for all pictures

We look forward to receiving your revised manuscript.

Kind regards,

Christos E. Chadjichristos

Academic Editor

PLOS ONE

Reviewers' comments:

Reviewer's Responses to Questions

**Comments to the Author**

1. If the authors have adequately addressed your comments raised in a previous round of review and you feel that this manuscript is now acceptable for publication, you may indicate that here to bypass the “Comments to the Author” section, enter your conflict of interest statement in the “Confidential to Editor” section, and submit your "Accept" recommendation.

Reviewer #1: All comments have been addressed

2. Is the manuscript technically sound, and do the data support the conclusions?

Reviewer #1: Partly

3. Has the statistical analysis been performed appropriately and rigorously? 

Reviewer #1: N/A

4. Have the authors made all data underlying the findings in their manuscript fully available?

Reviewer #1: Yes

5. Is the manuscript presented in an intelligible fashion and written in standard English?

Reviewer #1: No

6. Review Comments to the Author

Reviewer #1: Overall, the authors have addressed the points raised by the reviewers and I appreciate their effort. However, few responses are not entirely convincing/comprehensive.

The potential toxicity of the high dose is discussed, but no data shown to support their statement. An “immune-related influence” is suggested but, again, no supporting data.

The inflammatory infiltrate quantification is now provided based on a histological score. Specific immunostaining of inflammatory cells would have been preferable and more representative.

The authors are also showing a reduction in inflammation by measuring TNF. Why did they only measure TNF?

Figure 3-4 and 5 please ensure that the magnification is the same for all panels. Scale barre are not provided for figure.

WB in Figure 7. I would have preferred to see the individual mice, not to pool them.

7. PLOS authors have the option to publish the peer review history of their article (what does this mean?). If published, this will include your full peer review and any attached files.

Reviewer #1: No

---

## [Author Response · Author response to Decision Letter 1]

29 Sep 2020

Editor comments 1:

Please provide immunostainings for inflammatory cells.

Answer 1: 

Thanks for your comments, we have finished the immunostaining of f4/80, marker of macrophage, and the semi-quantitation has been completed. 

The trends of macrophage infiltration in each group were similar to other pathological evaluations. 

Editor comments 2:

Please provide scale bars for all pictures.

Answer 2:

The magnification for all panels is the same, and we have added scale bars for all pathological pictures.

Reviewer comments 1:

The potential toxicity of the high dose is discussed, but no data shown to support their statement. An “immune-related influence” is suggested but, again, no supporting data.

The inflammatory infiltrate quantification is now provided based on a histological score. Specific immunostaining of inflammatory cells would have been preferable and more representative.

Answer 1: 

Thanks for your comments, we have finished the immunostaining of f4/80, marker of macrophage, and the semi-quantitation has been completed. 

The trends of macrophage infiltration in each group were similar to other pathological evaluations. 

Reviewer comments 2:

Figure 3-4 and 5 please ensure that the magnification is the same for all panels. Scale bars are not provided for figure.

Answer 2:

The magnification for all panels is the same, and we have added scale bars for all pathological pictures.

Reviewer comments 3:

WB in Figure 7. I would have preferred to see the individual mice, not to pool them.

Answer 3:

We have retested the individual mice of each group again. The image of one set samples have been picked from six sets.

---

## [Editor Report · Decision Letter 2]

8 Oct 2020

Utilizing Methylglyoxal and D-lactate in Urine to Evaluate Saikosaponin C Treatment in Mice with Accelerated Nephrotoxic Serum Nephritis

PONE-D-20-05825R2

Dear Dr. Shih-Ming Chen,

We’re pleased to inform you that your manuscript has been judged scientifically suitable for publication and will be formally accepted for publication once it meets all outstanding technical requirements.

Kind regards,

Christos E. Chadjichristos

Academic Editor

PLOS ONE

---

## [Editor Report · Acceptance letter]

12 Oct 2020

PONE-D-20-05825R2 

Utilizing methylglyoxal and D-lactate in urine to evaluate saikosaponin C treatment in mice with accelerated nephrotoxic serum nephritis 

Dear Dr. Chen:

I'm pleased to inform you that your manuscript has been deemed suitable for publication in PLOS ONE. Congratulations! Your manuscript is now with our production department. 

Kind regards, 

on behalf of

Dr. Christos E. Chadjichristos 

Academic Editor

PLOS ONE